Spectrum of tablet computer use by medical students and residents at an academic medical center

Robinson Robert rrobinson@siumed.edu
Department of Internal Medicine, Southern Illinois University School of Medicine , Springfield, IL , USA
Perry George
Electronic publication date: 2015 Jul 30
Publication date: 2015
Volume: 3
Electronic Location ID: e1133
Received 2015 May 22; Accepted 2015 Jul 6
Copyright: © 2015 Robinson
Copyright year: 2015
Copyright holder: Robinson
License: This is an open access article distributed under the terms of the Creative Commons Attribution License, which permits unrestricted use, distribution, reproduction and adaptation in any medium and for any purpose provided that it is properly attributed. For attribution, the original author(s), title, publication source (PeerJ) and either DOI or URL of the article must be cited.
License URL: https://creativecommons.org/licenses/by/4.0/

Keywords: Medical students, Resident physician, Physician, Medical education, Tablet computer

Funding: The author received no funding for this work.

==============================
Introduction. The value of tablet computer use in medical education is an area of considerable interest, with preliminary investigations showing that the majority of medical trainees feel that tablet computers added value to the curriculum. This study investigated potential differences in tablet computer use between medical students and resident physicians.

Materials & Methods. Data collection for this survey was accomplished with an anonymous online questionnaire shared with the medical students and residents at Southern Illinois University School of Medicine (SIU-SOM) in July and August of 2012.

Results. There were 76 medical student responses (26% response rate) and 66 resident/fellow responses to this survey (21% response rate). Residents/fellows were more likely to use tablet computers several times daily than medical students (32% vs. 20%, p = 0.035). The most common reported uses were for accessing medical reference applications (46%), e-Books (45%), and board study (32%). Residents were more likely than students to use a tablet computer to access an electronic medical record (41% vs. 21%, p = 0.010), review radiology images (27% vs. 12%, p = 0.019), and enter patient care orders (26% vs. 3%, p < 0.001).

Discussion. This study shows a high prevalence and frequency of tablet computer use among physicians in training at this academic medical center. Most residents and students use tablet computers to access medical references, e-Books, and to study for board exams. Residents were more likely to use tablet computers to complete clinical tasks.

Conclusions. Tablet computer use among medical students and resident physicians was common in this survey. All learners used tablet computers for point of care references and board study. Resident physicians were more likely to use tablet computers to access the EMR, enter patient care orders, and review radiology studies. This difference is likely due to the differing educational and professional demands placed on resident physicians. Further study is needed better understand how tablet computers and other mobile devices may assist in medical education and patient care.

Introduction

Tablet computers such as the Apple iPad and Kindle Fire are extraordinarily popular with the general public and physicians. These tablet computers generally have wireless networking capability and the ability to be customized by installing user selected “apps.”

Apps are self-contained software applications with a diverse array of purposes ranging from entertainment to medical decision support. Over 1,600,000 apps are available for the Apple iOS platform (the operating system for the iPhone and iPad), with over 34,000 categorized as medical apps (PocketGamer, 2015). Medical apps typically cost less than $5 (Robinson & Burk, 2012), and generally work on smartphones and tablet computers. The essential app for many physicians is their electronic medical record (EMR).

Ease of use and large screens make tablet computers a natural fit for EMR access, computerized physician order entry (CPOE), and radiology image review. As many as one-third of physicians in the United States use tablet computers in clinical settings, with 14–35% using these mobile devices to access an EMR (Robinson & Burk, 2012; Sclafani, Tirrell & Franko, 2013). Accessing an EMR via tablet computers can decrease the total time physicians spend logged into workstations while on duty (Horng et al., 2012), improve the efficiency of inpatient medical care by facilitating earlier order entry (Patel et al., 2012), and appears to be preferred over traditional workstations in a hospital setting (Lehnbom et al., 2014). An observational study showed that tablet computers decrease inpatient data management time while increasing the time physicians spend interacting directly with patients (Fleischmann et al., 2015). These results are supported by systematic reviews that indicate tablet computer use can result in improved documentation, medical decision making, and physician efficiency (Mickan et al., 2013; Mickan et al., 2014). In addition, patients embrace mobile technology and report tablet computer use by physicians as a positive aspect of their medical care (Strayer et al., 2010). These factors are likely to fuel even greater adoption of tablet computers by physicians.

The value of tablet computer use in medical education is an area of considerable interest, with some medical schools integrating tablet computers into their preclinical curriculum (Dolan, 2011). Preliminary investigations shows that the majority of students at a medical school felt that tablet computers were a positive addition that added value to the preclinical curriculum (George et al., 2013), and another medical school reports improved United States Medical Licensing Exam (USMLE) test scores after integration of tablet computers into the curriculum (Comstock, 2013). Studies of tablet computer use by medical trainees in the United States showed that point of care references (i.e., drug guides), board exam study resources, curricular materials, and EMR data were the most common types of medical resources used by medical students and residents on tablet computers during clinical rotations (Sclafani, Tirrell & Franko, 2013; Robinson & Burk, 2013; Nuss et al., 2014; Archibald et al., 2014).

This study explores differences in tablet computer use between medical students and resident physicians at the same institution in hopes of providing insight into how these devices influence medical care and education. The working hypothesis was that tablet computer use and medical app use would substantially differ between medical students and resident physicians.

Materials & Methods

Data collection for this survey was accomplished with an online questionnaire shared with the medical students and residents at Southern Illinois University School of Medicine (SIU-SOM) in July and August of 2012. The SIU-SOM is located in Springfield, Illinois and had an enrollment of 298 medical students and 314 residents and fellows at the time of this survey. Satellite training sites in Carbondale, Decatur, and Quincy, Illinois were also included. Training sites include 5 hospitals and many outpatient clinics in central and southern Illinois.

This anonymous survey was approved by the Springfield Committee for Research Involving Human Subjects (SCRIHS), the local institutional review board.

This survey asked respondents about tablet computer use, type of tablet computer (iPad, Android, other), medical app use, and frequency of medical app use. A scale for the frequency of medical app use was used. This scale of frequency included several times daily, weekly, monthly, and never. “Never” was included because it was possible that students might only use tablet computers for personal purposes (email, games, etc.) and not use applications designed for medical purposes.

Qualitative variables were compared using Pearson chi2 or Fisher’s exact test and reported as frequency (%), and p values less than 0.05 were considered statistically significant. SPSS version 17.0 was used for data analysis.

An inactive copy of the survey instrument can be accessed at: http://goo.gl/wn5QU.

SIU-SOM or any affiliated hospitals do not require or issue tablet computers to students or residents. The training hospitals at SIU-SOM and the faculty outpatient practice have electronic medical records that are accessible via tablet computers in accordance with SIU-SOM and hospital policies.

Results

There were 76 medical student responses (26% response rate) and 66 resident/fellow responses to this survey (21% response rate). Slightly over 50% of respondents used a tablet computer, with the Apple iPad being the most popular type of tablet computer used (Table 1). Tablet computer based use of medical apps one or more times daily was reported by 40% of respondents (Table 2). Residents/fellows were more likely to use tablet computers several times daily for medical apps than medical students (32% vs. 20%, p = 0.035). A high percentage of medical students (54%) and residents (50%) report never using their tablet computers to access medical applications.

Table 1 Tablet computer use by physicians in training.

	Medical students	Residents/fellows	
Total respondents	76	66	
Use a tablet computer	39 (51%)	33 (50%)	
Use an iPad	35 (46%)	27 (41%)	
Use an android tablet	4 (5%)	6 (9%)	

Table 2 Frequency of medical app use by physicians in training.

	Medical students	Residents/fellows	
Total respondents	76	66	
Use medical apps	35 (46%)	33 (50%)	
Several times daily	15 (20%)	21 (32%)	
Daily	10 (13%)	11 (17%)	
Weekly	10 (13%)	1 (2%)	
Monthly	0 (0%)	0 (0%)	
Never	41 (54%)	33 (50%)	

Respondents were asked about how they use tablet computers (Fig. 1). The most common reported uses were for accessing medical reference applications, e-Books, and board study. Residents were more likely than students to use a tablet computer to access an electronic medical record (41% vs. 21%, p = 0.010), review radiology images (27% vs. 12%, p = 0.019), and enter patient care orders (26% vs. 3%, p < 0.001).

Figure 1 Tablet computers use by stage of medical education.

Discussion

This study shows a high prevalence and frequency of tablet computer use, primarily in the form of iPads, among physicians in training at this academic medical center. Most residents and students use tablet computers to access medical references, e-Books, and to study for board exams. The high frequency of e-Book and other educational material use on tablet computers suggest that this may be an important avenue for medical educators to investigate.

These results show a higher rate of resident tablet computer use (50% vs. 13–19%) in clinical settings and a higher rate of EMR access (41% vs. 14%) than reported in a similar study investigating attending, resident and fellow tablet computer use in other institutions (Sclafani, Tirrell & Franko, 2013). Rates of tablet computer use by medical students are similar to the results in a previously published nationwide survey (Robinson & Burk, 2013). These differences in tablet computer utilization between these studies may be related to information needs at different levels of training, the level of institutional support for tablet computers, or other factors such as survey design.

Residents were more likely to use tablet computers for direct patient care such as accessing an EMR, reviewing radiographs, and CPOE. This suggests that computers become integrated into the workflow of residents at SIU-SOM, which is not unexpected given reports of improved resident efficiency and increased time in direct patient care with tablet computer use (Horng et al., 2012; Patel et al., 2012). This also is likely a reflection of the transition of mobile computing needs from that of a student to that of a practitioner.

Tablet computer use is likely complementary to high rates of smartphone use for many medical tasks by medical students and residents (Franko & Tirrell, 2012; Payne, Wharrad & Watts, 2012). Further investigation is needed to determine the preferred platform (phone vs. tablet) for access of healthcare information.

One concern regarding the prevalence of a “bring your own” tablet computer for accessing healthcare information is the relative ease in which information can be shared within a tablet computer. Text can be cut and pasted, images saved, and information shared through other applications. This is because less than half of third year medical students think sharing patient photographs via social media is “definitely a privacy concern” (Whipple, Allgood & Larue, 2012) and 1% of medical student and resident Facebook profiles include identifiable photographs of patients in healthcare settings (Thompson et al., 2011). Less than half of medical students password-protect patient data on their mobile devices (Whipple, Allgood & Larue, 2012). Medical schools must implement policies and procedures sensitive to technological improvements that protect patient privacy in the new mobile world.

As a single center study, the results of this survey may not be generalizable. Local factors, such as the availability of EMR access via a tablet computer, are likely to have had significant effects on the patterns of tablet computer use in this study. Additionally, this survey may have selection bias given the voluntary nature of this survey. Despite these limitations, this survey shows widespread adoption and daily clinical use of tablet computers in an academic medical center by medical students and residents.

Conclusions

Tablet computer use among medical students and resident physicians was common in this survey. All learners used tablet computers for point of care references and board study. Resident physicians were more likely to use tablet computers to access the EMR, enter patient care orders, and review radiology studies. This difference is likely due to the differing educational and professional demands placed on resident physicians. Further study is needed better understand when and how tablet computers and other mobile devices may assist in medical education and patient care.

Supplemental Information

Supplemental Information 1 Raw Dataset

Click here for additional data file.

Additional Information and Declarations

Competing Interests

Author Contributions

Human Ethics

The author declares there are no competing interests.

Robert Robinson conceived and designed the experiments, performed the experiments, analyzed the data, contributed reagents/materials/analysis tools, wrote the paper, prepared figures and/or tables, reviewed drafts of the paper.

The following information was supplied relating to ethical approvals (i.e., approving body and any reference numbers):

Springfield Committee for Research Involving Human Subjects (SCRIHS) gave a Notification of Exemption Approval for this study, Reference number: 000838.

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
