# Peer review of "Spectrum of tablet computer use by medical students and residents at an academic medical center"

_PeerJ, doi:10.7717/peerj.1133_

## Round 0.1 · original submission · Minor Revisions

The reviewers have outlined several issues that should be addressed prior to publication

Reviewer 1 ·

Basic reporting

The article was clearly written and was an appropriate length and the material presented was relevant to the claims being made by the authors.

Experimental design

The design was simple and straightforward but appropriate to the study. An expanded explanation of how the statistics were calculated may be useful.

Validity of the findings

The significance of the data is propelling. Possible expansion of the discussion and conclusions may be helpful to understand why this study is so important and deserves publication

Additional comments

Article was easy to understand and straightforward. Would be helpful to expand why this information is important.

Reviewer 2 ·

Basic reporting

1. Line 24. Should not start a sentence with an abbreviation (…EMR…).

2. What is CPOE in Figure 1?

Experimental design

1. Line 71. What is meant by daily? More than once a day, twice, ten times a day?

2. Is it just one hospital that is part of this study? If so, does the hospital (or if more than one site, hospitals) supply the tablets for the residents? If so, this would be the likely reason for use. If they had to purchase their own there might be problems with access to data bases such as Epic and may have HIPPA issues as well. It is indicated in Line 63 that SIU-SOM does not supply, but the hospital may. If so, this may skew the data towards resident use.

Validity of the findings

1. Some comment should be made about the “Never” use in Table 2.

2. Some comment should be made concerning mobile cell phones as an alternative to the tablet.

3. Medical student use may be more limited to using tablets due to time constraints of school hours compared to resident use as part of their routine encounters with patients. What about after hour use versus use during the working/class day?

Additional comments

This manuscript reports on the use of tablet computer by medical students versus medical residents at Southern Illinois School of Medicine. The authors conclude that tablet use is a common practice in medical education and that medical residents are more likely to use their tablets computers on a daily basis compared to medical students. This is most likely due to clinical activities of medical residents requiring the need of a portable computer. This study builds upon prior work done by the author and others, but the major finding is the comparison of medical students versus residents use.

This manuscript adds to the literature on computer use and in particular, mobile devices, for use in medical education and patient care. There are some minor concerns with this study that need to be addressed.

---

## Round 0.2 · accepted · Accept

Thank you for addressing the reviewer isssues